# Assessment and Scientific Progresses in the Analysis of Olfactory Evoked Potentials

**DOI:** 10.3390/bioengineering9060252

**Published:** 2022-06-12

**Authors:** Pasquale Arpaia, Andrea Cataldo, Sabatina Criscuolo, Egidio De Benedetto, Antonio Masciullo, Raissa Schiavoni

**Affiliations:** 1Department of Information Technology and Electrical Engineering (DIETI), University of Naples Federico II, 80125 Napoli, Italy; pasquale.arpaia@unina.it (P.A.); sabatina.criscuolo@unina.it (S.C.); egidio.debenedetto@unina.it (E.D.B.); 2Department of Engineering for Innovation, University of Salento, 73100 Lecce, Italy; antonio.masciullo@unisalento.it (A.M.); raissa.schiavoni@unisalento.it (R.S.)

**Keywords:** olfactory dysfunction, smell, anosmia, hyposmia, olfactory evoked potentials, ERPs, OERPs, brain, electroencephalography, grand averaging

## Abstract

The human sense of smell is important for many vital functions, but with the current state of the art, there is a lack of objective and non-invasive methods for smell disorder diagnostics. In recent years, increasing attention is being paid to olfactory event-related potentials (OERPs) of the brain, as a viable tool for the objective assessment of olfactory dysfunctions. The aim of this review is to describe the main features of OERPs signals, the most widely used recording and processing techniques, and the scientific progress and relevance in the use of OERPs in many important application fields. In particular, the innovative role of OERPs is exploited in olfactory disorders that can influence emotions and personality or can be potential indicators of the onset or progression of neurological disorders. For all these reasons, this review presents and analyzes the latest scientific results and future challenges in the use of OERPs signals as an attractive solution for the objective monitoring technique of olfactory disorders.

## 1. Introduction

Olfactory function is one of the earliest phylogenetic senses developed in the history of human evolution [1], mainly for survival reasons related to the location and hunt of food, dangers and enemies [2]. The sense of smell plays a major role in our daily lives because it represents one of the most important ways of interacting with the environment, through the ability to acquire and interpret nearby chemical signals. This capacity is innate since birth, as a newborn can recognize his mother only from the smell, especially through the odor emanated by the mother’s milk [3]. Besides the prehistoric and primordial aspects, odor perception has an enormous impact on human development; in fact, recent studies have demonstrated that the cortical areas related to olfactory sense have significant connections with memory, speech and neurovegetative areas. In addition, the role of olfactory signals appears to influence tastes, personality and cognitive functions, such as attentional processes [4], generating memories and emotions and modulating behavior and interpersonal relationships. In this regard, researchers are investigating the characterization of the chemicals signals that cause happiness and fear [5]; the goal is to develop an odor delivery system able to drive the social–emotional state of the subjects, in both a social and clinical scenario, for the treatment of anxiety and depression.

Considering the impact of smell recognition in human life, a partial or total loss of smell (hyposmia or anosmia) has a profoundly adverse effect on the overall quality of life [6], compromising normal living habits [7] and making changes in appetite and mood [2,8]. Another problem arising from this dysfunction is the possible exposure of the subject to safety risks [9], resulting from the lack of perception of gases or harmful agents. This eventuality increases significantly especially in old age, as advancing years often cause the deterioration of olfactory function. [10]. The causes of anosmia and hyposmia have long been investigated, establishing that olfactory disorders usually occur after sinonasal diseases [11], upper respiratory viral infection [12], continuous exposure to toxic agents [13,14] and skull fractures [15]. It is also important to underline that a decrease in smell can also be a symptom of particular silent pathologies, e.g., neuro-degenerative diseases [16,17], such as Parkinson’s disease, Alzheimer’s disease, and mental health issues, e.g., anxiety and depression [7]. This means that olfactory deficit can be used as a marker for the early diagnosis of neurological pathologies, allowing the beginning of the therapy with timing and better life perspectives. Recently, the problem of the loss of the ability to perceive odors has become more critical, as it is one of the first symptoms of COVID-19 infection [18], which may persist even after the patient has fully recovered, causing serious discomfort [19,20]. Another aspect, which is no less important, is the study of smell and its relationship with the brain in sensory marketing and communication activities, in order to recall flavors, fragrances and good memories in the consumer’s mind [21].

On the basis of these considerations, the possibility of having available reliable techniques for an objective assessment of olfactory function is crucial. Over the years, several solutions for investigating smell capabilities have been proposed; nevertheless, qualitative orthonasal olfactory tests are still the most used. Clinical tests of olfactory function have been developed since 1980 under the name of the University of Pennsylvania Smell Identification Tests (UPSIT) [22], which release odors through the use of a pencil on a sheet of paper. The patient is required to detect the odor from four different choices and to answer some questions. Similarly, in 1997, the Sniffin’ Sticks test [23] was developed by Hummel, as a new method to identify the patient’s olfactory response based on a pen-like odor diffuser. However, tests of this type focus on a psychological method of investigation based on the subject’s ability to identify odors, presented by a paper strip or felt pen, over a threshold. Although this kind of test has been widely adopted over the years because of its low cost, practicality, efficacy and relative reliability [24], this method is based on the subjective response of the person under examination. Therefore, psychological tests suffer from a lack of objectivity of diagnosis, resulting in being inappropriate in the case of children, or in patients who are unable to participate appropriately in tests, or in a medico-legal perspective [25,26], where the patient’s honest participation cannot be taken for granted. These limitations have encouraged scientists to explore objective methods to assess olfactory performance, focusing on the use of the olfactory event-related potentials (OERPs), developed since the 1960s [27]. This method is based on the recording of electroencephalography, by electrodes on the scalp, of responses of brain activity after olfactory stimulus presentation. As a matter of fact, the assessment of smell function with OERPs could provide more reliable and objective information because it is less dependent on patient collaboration and it is based on the brain’s spontaneous response to a stimulus. In addition, this method is attractive for the ability to analyze the neural signals through a safe and non-invasive approach, simply by using a helmet equipped with electrodes in contact with the scalp. However, in spite of the technical progress, considerable improvements are still needed for the affirmation of OERP use in clinical and diagnostic fields, especially because of problems related to noise and because of the differentiation of olfactory responses from cognitive processes. In this regard, different analysis and signal processing techniques are used.

On such a basis, taking into account the great potential and the complexity of the topic, the aim of this review is to gather the latest and brightest results regarding the use of olfactory evoked potentials, focusing mainly on the technological and engineering aspects. In fact, for the state of the art, all studies that focused on the analysis of olfactory dysfunction appear to be purely medical and are often not exclusively focused on OERPs [16,24,28,29]. Undoubtedly, such studies are of fundamental interest as a guide and support for the comprehension of chemical–physical processes related to smell and brain mechanisms, but it is necessary to reorganize and order the latest technological developments as a basis to take further steps forward for their consolidation, especially in healthcare applications. For this purpose, this paper is organized as follows. Section 2 describes the basic theoretical background behind the OERPs signals. Section 3 shows the methodology used to record, process and analyze OERPs. Section 4 presents the main applications in which OERP signals are used. Section 5 introduces the challenges and future work. Finally, the conclusions are outlined in Section 6.

## 2. Olfactory Event Related Potentials

The OERPs, as the other evoked potential, are poly-phasic signals generated in the brain in response to the activity of the cerebral cortex neurons, producing electro-magnetic fields as an electrophysiological response to specific stimuli [30], in this case olfactory stimuli. OERPs can be reliably measured using electroencephalography (EEG), which measures and records instantaneous electrical potential differences between different areas of the brain, through the placement of electrodes on the scalp. Generally, an OERP consists of a waveform characterized by a series of positive or negative components, followed by a late positive component of higher amplitude that is a positive-going event-related brain potential [31]. Figure 1 shows three examples of OERP. The OERP signal is characterized by three main parameters:Latency: time interval between the stimulus onset (fixed to 0 ms) and the point of maximum value (peak) of the component;Topography: position on the cranial surface where the maximum amplitude of the component can be registered, thus allowing identification of which cortical area is active following a particular stimulus;Amplitude: vertical distance measured from the baseline (fixed to 0 μV) to the maximum peak.

Furthermore, as shown in Figure 1, there are different event-related components as the positive peaks named P100, P200, and P300, and the negative peaks as N100 and N200. Their names, usually, indicate wave polarity and absolute latency time after stimulation. Thus, these waves occur at about 100 ms, 200 ms, or 300 ms after the stimulus. However, this absolute latency time is often an approximation, and it can often vary. For this reason, what matters is the latency order, which is why expressions such as P1, N1, P2 and so on, are preferred [32,33]. Analyzing in more detail Figure 1, firstly, there is a small positive response P1, which is not always defined, followed by a large negative response N1 (amplitude: −3 μV to −10 μV) and, finally, a large wave P3 (amplitude: +5 μV to +20 μV) [34]. The shape and latency of these event-related potentials allow the detection of sense deficits and the quantification of their severity; moreover, the size of the trace that reproduces the generated potentials changes with stimulus concentration and adaptation phenomenon [31,35]. In more detail, the initial components of the OERP, P1, N1, P2, and N2 are called exogenous sensory components and are related to odor threshold and odor identification [36,37], in particular, the first negativity of the evoked potential corresponds to the nature of the stimulus [34]. On the other hand, the P3 component relates to endogenous stimulus processing and is therefore related to the cognitive processing speed and the ability to evaluate and classify a stimulus [38,39]. When an odorant molecule triggers olfactory cells, after generating a series of alternating positive and negative waves (exogenous components), a later potential (P300) can be measured with electrodes placed over the center of the scalp in the central and parietal areas [26]. This P300 response is involuntary and, consequently, it is considered more objective than that obtained through psychophysical tests [34]. For this reason, identifying the P300 wave and its characteristics is of paramount importance to obtain information about olfactory function.

With regard to the type of stimulus and to the electrode placement, another important aspect to consider relates to variations in the specific EEG frequency band, compared to the pre-stimulation condition. In this regard, different studies have investigated the effects of the olfactory stimulus on different EEG frequency bands. Although there is an evident alteration of the EEG activity, the frequency band involved is often varied in several studies, likely due to the different methodologies considered. Indeed, in [40], it is shown that the theta frequency band has statistically significant reduction, in agreement with previous studies [41,42,43]. On the other hand, more recent studies based on time–frequency analysis of the EEG signal consider the gamma band as being associated with smell [44,45].

### Reliability of OERPs

As discussed in the previous sections, OERP appears to be a more objective diagnostic tool than standard psychophysical testing [46,47]. Nevertheless, in comparison to visual and auditory evoked potentials, widely used very early, OERPs were recorded reliably only at the beginning of the 1980s [48], due to the difficulty in presenting a controlled odorous stimulus, without having a gustatory stimulation. Since then, OERPs have enjoyed increasing popularity in the study of smell disorders. However, the reliability of this measurement strategy was examined only starting from the early 2000s, by test–retest over a 4-week interval with healthy participants [49]. More specifically, this study showed that P2 is related to stimulus characteristics and correlates with olfactory threshold tests [37]; it may, therefore, serve as an indicator in most clinical and experimental studies. Likewise, olfactory P3, having a high level of stability in relation to its temporal occurrence, could suggest the use of P3 latency as a consistent measure of central olfactory function.

Thereafter, the reliability of olfactory ERPs in investigating olfactory function was demonstrated in the literature by various studies correlating the electrophysiological assessment of olfactory function with concomitant psychophysical assessments. More specifically, OERPs signal are never detected in patients diagnosed with anosmia using psychophysical tests (e.g., Sniffin’ Sticks) but could be present in some hyposmic patients [26,50] but with different characteristics from those of the control. On the other hand, as was reported in the Rombaux study [50], in patients with normal olfactory function, the absence of OERP is probably related to the technical problem, for instance EEG artifacts. Certainly, the odd results of OERP presence in a patient diagnosed as anosmic with psychophysical tests could mean that the patient lied probably for a secondary gain. Table 1 summarizes some of the results from [50].

Similarly, in [51], the OERP responses of patients with olfactory impairment, assessed with psychophysical tests, and healthy controls to odor concentration was evaluated. It was noted that as the stimulus concentration increases, OERP amplitudes increased, and the latencies shortened. Additionally, the analysis on the control group and the patient group showed higher OERP amplitudes and shorter latencies in healthy subjects compared with patients. More recently, in [52], comparing the perception of odors with traditional olfactometry and the amplitude of evoked potentials, the correlations between the two types of tests, subjective and objective, were evaluated in patients with post-traumatic anosmia. Results in [52] were in good agreement with the previous studies. The authors demonstrated, also through the support of subjective tests, a marked reduction in the OERP amplitude or the total absence of the OERP component, possibly associated to a serious olfactory dysfunction. Contrarily, in cases where subjective tests reported impairment in smell function but the OERP, resulting in reaction to the odor, was present, it can be concluded that subjects feigned anosmia or cognitive impairment.Finally, with the COVID-19 pandemic, a preliminary study was conducted on the prognostic value of the wave ERPs parameters [53] and thus their usefulness as predictors of olfactory recovery after infection. More specifically, a comparison between subjective test and OERPs result showed that patients with the shortest latencies of N1 had a higher probability of recovery than patients with the longest latencies of N1.

## 3. Registration and Pre-Processing

After the introduction to event-related olfactory potentials in Section 2, this section presents the methodology used to record the OERPs and to analyze them.

### 3.1. ERP Experimental Setup

As mentioned in Section 2, the midscalp region is the area where OERPs are generally recorded, while the largest peak is observed at the central and parietal areas [37]. Typically, the electrodes are placed at the Fz, Cz and Pz positions, along the midline of the helmet, according to the International 10–20 system [54], shown in Figure 2. Besides these three, other electrodes are placed to work as the reference and ground, or for assessing motor artifacts, in particular, related to eye blinks.

In Ref. [37], which focused on olfactory evaluation in the elderly and younger people, brain signals were acquired by non-polarizing Ag/AgCl active electrodes on the midline, with reference to the right mastoid process, and with a ground electrode to the left mastoid process. Furthermore, two electrodes were placed at the left eye to monitor ocular artifacts. More in detail, the brain activity was recorded, filtered (0.1–30 Hz bandpass) and digitized using the bio-logic device [55]. To stimulate the OERP signal, a specific olfactometer made in 1993 was used [56], delivering an amyl acetate odor.

A similar configuration for the experimental setup is found in the studies carried out by Morgan et al. [36], by Covington et al. [57], and by Thesen [49]. The major difference is that gold-plated electrodes were used to record EEG activity.

Although the position for the three main electrodes (Fz, Cz, Pz) was kept the same throughout the years, it is worth noting that many studies have used more electrodes placed on the scalp. For example, Wang et al. [35] focused on finding a correlation between the psychophysical and psychological responses to olfactory stimulation: in this case, the electrodes were located at Fz, Cz, Pz, T3, C3, C4 and T4, referred to as A1 (left ear lobe), and a ground electrode was placed on the frontal area. In detail, the acquired data were digitized at 100 Hz and then a low-pass filter (35 Hz) and band-stop filter (50 Hz) were used. The ERP was evoked by an olfactometer developed in 1999 [58], and the used odor was isoamyl acetate. Positions C3 and C4 were also considered by Lötsch and Hummel [26], who monitored blink artifacts with an electrode in position Fp2 (Figure 2). As a matter of fact, in this study, EEG was acquired from the Cz, C3, C4, Fz, and Pz positions, referenced to linked earlobes (A1 and A2). The EEG signal was digitally recorded at a frequency of 250 Hz. The OERP was stimulated by using phenylethyl alcohol. Subsequently, Guo et al. [53] also considered positions C3 and C4. In this study, EEG activity was recorded at a 250 Hz sampling rate from the Fz, Cz, Pz, C3, and C4 positions with reference electrodes in the left and right earlobes (A1 and A2). The ERP signal was evoked by phenethyl alcohol.

As we can see, the experimental setup generally involves electrodes placed in the central area of the scalp, as the signals recorded in this area are more related to the olfactory stimulus [40].

### 3.2. ERP Processing Techniques

As mentioned in Section 2, the measurement of the EEG signal must take into account the presence of noise or the effect of interference.

In fact, although EEG instrumentation is designed to record brain activity, the recorded trace consists of not only the desired signal, but also of undesired components called artifacts, which can be of physiological or non-physiological origin [59]. The first category includes signals that originate from the patient and are usually more complicated to detect and eliminate, such as eye, muscle, heart, or skin artifacts. Instead, the second category (non-physiological) includes electrode artifacts typically caused by non-optimal application to the scalp) and instrumentation or network artifacts. In order to eliminate artifacts, the independent component analysis (ICA) technique [60,61] is often adopted to dissociate the original multivariate signal into the linear combination of multiple mutually independent sources. After this operation, the OERPs extraction is still not easy: in fact, background EEG is also assimilated to noise, as an unwanted signal superimposed on the component of interest. As a result, the signal-to-noise ratio (SNR) between ERPs and EEGs is rather low due to the EEG having higher amplitude than the ERP [62,63] and having common frequency content. As a matter of fact, because it is impossible to set an “optimal” cut-off frequency and to isolate the ERP signal from the noise background EEG in the frequency domain, the easiest classical technique used is the averaging technique (also called grand averaging). In particular, this technique is based on the calculation of the average of *N* temporal segments of interest of the EEG, called target epochs [64]. These segments coincide with the parts of the trace that follow the same olfactory stimulus, so it is assumed that they contain the same type of evoked potential. The average $y_{m}\left(t\right)$ of the acquired *N* sweep targets $y_{i}\left(t\right)$, i=1,...,N is
(1)$$\begin{array}{c} y_{m}\left(t\right)=\frac{1}{N}\sum_{i=1}^{N}y_{i}\left(t\right)=\\ =\frac{1}{N}\sum_{i=1}^{N}\left(s_{i}\left(t\right)+v_{i}\left(t\right)\right)=s\left(t\right)+\frac{1}{N}\sum_{i=1}^{N}v_{i}\left(t\right) \end{array}$$
where $s_{i}\left(t\right)$ is the OERP in the *i*-th epochs target and $v_{i}\left(t\right)$ is the background EEG noise superimposed to the ERP of interest (according to an additive model), and s(t) is the deterministic ERP, assumed to be the same in all epochs. Because the amplitude of the ERP is small compared to background noise, artifacts and other disturbances, averaging becomes necessary, as it allows to reduce and ideally cancel the effect of the latter, enhancing instead the common components, as it is possible to note from Equation (Equation 1). The overall process is shown in Figure 3, which reports the grand average procedure for a specific example channel. The latency of a given OERP tends to be approximately constant or with small phase variations, whereas the fluctuations of the background EEG (noise in ERP analysis) are completely unrelated and unsynchronized. Hence, the evoked response, $y_{m}\left(t\right)$, is cleaned from the non-phase random oscillations, elided by the simple averaging process represented by the last addend of Equation (Equation 1). Although the averaging technique is the most widely used and established approach, it has some drawbacks [64,65,66]. In fact, the following aspects are not considered by the averaging technique:Variability of the evoked potential: amplitude and latency can vary independently of each other from epochs to epochs. Actually, significant latency jitter can result in a severely distorted and amplitude-reduced ERP average.Non-stationarity of the EEG: the basic assumption of averaging is that the background EEG is a random, null averaged, uncorrelated and stationary signal during the recording of the *N* epochs, but in reality EEG is assimilable to a stochastic model only for short stretches.No a priori knowledge about the relationship between EEG and ERP is exploited, so it is necessary to have a large number of sweeps before resorting to the grand averaging algorithm.

## 4. Application Fields

As explained in Section 1, olfactory function evaluation has become crucial through the years in several application fields, in which extensive research activity has been dedicated to explore the OERPs signals’ potential. In particular, the most important and interesting areas of applications are selected and reported in Figure 4, and the related most important and promising results are described in the following subsections. In all reported works, EEG datasets were obtained through EEG recordings on volunteer subjects who decided to participate in the trials. In more detail, the number of participants was typically more than 10 in each study, and they included both normal young and normal elderly people and subjects with neurodegenerative diseases. The sick people were mainly Alzheimer’s, Parkinson’s, or multiple sclerosis patients diagnosed by neurologists, while for the normal controls, olfactory and mental health assessments were made. Olfactory stimulation was accomplished by means of an olfactometer capable of delivering different kind of perfumes, and recording was performed through a helmet with a variable number of electrodes on the scalp. However, case analysis, operational methods and adopted processing techniques for each study are reported in detail. Finally, as a common point, for an accurate statistical analysis of the large amount of data and results, univariate and multivariate analysis of variance (ANOVA and MANOVA) [67] were generally used in these studies.

### 4.1. Olfactory Function Decline in Elderly People

The examination of age-related decline in olfactory function is one of reasons to encourage research in the OERPs study. This relationship, indeed, was investigated since the 1970s [68] and it was demonstrated that aging impairs the ability to identify odors due to multiple factors [69], such as altered nasal engorgement, progressive damage to olfactory system due to viral or environmental aspects. This problem is more serious than expected, especially considering the relation between olfactory dysfunction and higher mortality in the elderly [70]. In particular, a smell deficit in older adults may adversely affect nutrition, because the appetite declines, compromising the nutritional status [2]. In addition, risks from gas poisoning become more likely for seniors living alone if they are unable to detect the harmful agent in time [9]. As a matter of fact, less than a quarter of those with smell diseases know about their problem until they are tested, and this unawareness can be more dangerous for older persons, leading, in some cases, to early mortality [71]. This scenario leads to the urgency of a reliable olfactory test and, in this regard, the use of OERPs plays a fundamental role when psychophysical tests cannot provide reliable information [36,37]. Several studies pointed out that older adults produce smaller OERP amplitudes and longer latencies when compared to young adults [57,72]. This phenomenon is shown in Figure 5, which reports some results from [57]. In particular, Figure 5 provides three examples of OERP signals acquired from young and older people. It is noted that the OERP amplitude of an older subject is smaller, and the peak latency is longer than that of a young subject. These results suggest a reduced or absent reaction by the nervous system of elderly people to odors, making this approach an accurate assessment of age-related odor perception. In the same study [57], an ANOVA analysis was used to compare variances across the means (or average) of different groups. In particular, an average threshold value was computed and submitted to a two-factor (age x gender) ANOVA: (i) females exhibited lower thresholds (higher sensitivity) than males (*p* < 0.001); and (ii) as expected, young adults showed lower thresholds (higher sensitivity) than older adults (*p* < 0.08). These results show that the OERPs analysis allows to identify a decrease in olfactory sensitivity related, besides age, also to gender. The same result was found in [36], demonstrating that older males reported greater olfactory deficits than older females, suggesting a greater slowing of the elderly male brain than elderly females in response to olfactory stimulation.

### 4.2. Diagnostic in Neuro-Degenerative and Neuro-Psychiatric Diseases

Recently, smelling disorders have attracted considerable interest in the neurological and neuro-psychiatric field since olfactory dysfunctions are associated with the possible onset of Alzheimer’s disease (AD) and idiopathic Parkinson’s disease (PD) [73]. As is well known, AD is the most common form of dementia, with the loss of memory and other intellectual abilities, while PD is a neuro-degenerative disease characterized by a progressive loss of dopaminergic neurons in the substantia nigra [10], and involves, primarily, movement and balance control [74]. It is worth noting that clinical signs of AD, such as neurofibrillar tangles, were found in the olfactory bulb before the onset of typical symptoms [75,76]. In addition, anomalies in OERPs signals have been identified in individuals with the ApoE $\varepsilon_{4}$ gene allele [77,78,79], which is associated with an increased risk of the late onset of AD. Similarly, some PD patients have impaired olfactory sense several years before motor symptoms appear [10], as documented since 1975 [80]. In particular, the decline in olfactory function occurs in approximately 90% of patients with early stage PD [81], as has been investigated by studies since 1995 [82], and, more importantly, olfactory dysfunction is among the first non-motor features of PD [10].

These results reveal the close connection between smell disorders and risk of neuro-degenerative disease, suggesting that olfactory deficits may be early indicators and predictors of neuro-degenerative diseases, helping to identify subjects at risk. In this regard, the use of OERPs has gained momentum, as it allows to observe changes in olfactory function in an objective way [84,85].

In this context as well, several OERP studies have observed latency delays after olfactory stimulation in AD and PD. In particular, based on the study [83], Figure 6 shows the comparison between OERP from normal control subject and for one AD patient; it can be noticed that AD signals exhibit longer latencies (components shifted to the right) and lower peak amplitudes. More in detail, different components (olfactory N1, P2, N2, and P3 amplitude and latency) were analyzed, using a separate repeated measures MANOVA, considering different electrode sites (Fz, Cz, and Pz). This procedure showed significant differences between the OERP component of AD patients and the normal control, but also a significant effect of the electrode site.

As mentioned, olfactory dysfunction occurs also in patients with Parkinson’s disease. In particular, idiopathic Parkinson’s disease is related to latency delays in OERPs registration [84]. Conversely, in atypical Parkinson’s disease, the olfactory function is mostly preserved [86]. In light of this, analysis of olfactory function by using OERP could well reveal significant information of idiopathic Parkinson’s disease, allowing to distinguish from other neurodegenerative disorders, e.g., supranuclear palsy or corticobasal degeneration [86].

Furthermore, in addition to AD and PD diseases, also olfactory dysfunction in patients with schizophrenia [36] or multiple sclerosis (MS) [87] is becoming a topic of increasing interest. In schizophrenia cases, it appears that the major indicator is no longer the OERPs latency; on the contrary, schizophrenic subjects show different OERPs amplitudes compared to healthy individuals. Indeed, reduced N1 and P2 components have been found both in schizophrenics [88,89] and in unaffected close relatives of schizophrenics [90]. However, it should be mentioned that some studies have also pointed out differences in latency, although not very clearly [84]. As a matter of fact, in some studies, a longer latency was found for P2, while others reported reduced N1 and P2 latencies.

Finally, recent findings on the neurophysiopathology of multiple sclerosis reported the correlation with neurodegenerative processes of olfactory structures. Indeed, a study [85] was recently performed on 30 MS patients and 30 healthy subjects to objectively assess the smell function by using OERP. The results showed that MS patients had a mean value of OERP latency and amplitude significantly longer and lower, respectively, than the control group and, in some cases, the OERP component was totally absent. These results are summarized in Figure 7, showing that the parameter that varies most visibly is the amplitude of P2. In addition, the authors reported a connection between the presence or absence of evoked potential and the disability duration: it was noted that in patients with longer disease duration, there could be higher probability to not record OERP.

### 4.3. Analysis of Emotions for Informative Purposes

The sense of smell is also closely associated with behavior and emotion, thus, an olfactory dysfunction can affect them. Indeed, compared to healthy patients, anosmic subjects show significantly higher rates of depression and greater anxiety [7], as well as problems in social and family life [91]. Additionally, in this context, using correlated evoked potentials, the influence of a specific affective state on olfactory processing can be highlighted.

In particular, in [92], a study on 31 male subject showed that an emotional state of sadness, closely related to depression, is associated with a decrease in olfactory sensitivity. More specifically, longer latencies of N1 and P2 peaks, and lower amplitudes were observed in the presence of an unpleasant odor [92].

However, emotional information is often expressed by cross talk between sensory channels. In fact, OERPs were shown to regulate the recognition of facial expressions, reporting that unpleasant odors triggered faster recognition of facial expressions, especially fearful ones. More specifically, it was found that unpleasant food odors evoked a P100 with greater amplitude when compared to pleasant and neutral food odors. This promotes the rapid processing of facial expressions [93].

Olfactory emotions also play a key role in marketing and communication, prompting companies to pay more attention to the new frontier of the study of odors. In fact, fragrances influence our cognitive functions, stimulating different brain areas and, therefore, the olfactory-enhanced multimedia experience strengthens the perception of reality and the quality of the user’s experience [94], influencing the consumer’s purchase decision process. Again, the analysis of changes in EEG signal to different odors can be helpful in investigating a subject’s perception of an odor, and several studies have attempted to do this, although they are very different in terms of methods and results [47]. Furthermore, OERPs can be a useful tool to investigate physiological and psychological responses to odor intensities, adaptation and habituation to odors, which are fundamental aspects of the user experience. More in detail, it has been observed that odor concentration is generally correlated with the amplitude of OERPs and not with latency [35]. Despite all the advantages, however, OERPs are very sensitive to age, and the reaction time to stimuli also depends on gender and personality [95]. Therefore, extracting robust conclusions from OERPs regarding the emotional aspect of smell is still difficult.

## 5. Challenges and Future Goals

The study and the use of OERPs signals has achieved important results in different application fields, but further improvements are required to achieve increasingly important goals on several fronts.

### 5.1. Standardized Methods

As discussed in the previous sections, the measurement of the olfactory ability is fundamental for earlier diagnosis of neuro-degenerative and neuro-psychiatric diseases and for prevention in the elderly. Hence, the possibility of applying the OERPs technique in medical fields is of great interest. Nevertheless, the lack of standardized methods in acquired EEG associated with olfactory function prevents the reproducibility of the assessment. As a matter of fact, the main disadvantage of OERPs is the requirement of a stimulator able to provide a short synchronized and selective olfactory stimulus [50]. In addition to this, it is observed that the latency and amplitude of OERPs depends on several aspects, such as the properties of the stimulus (e.g., concentration) [35], the olfactory fatigue or habituation. The psychological conditions also affect ERPs; in particular, the P300 component reflects cognitive states. In more detail, the P300 component can also be caused by other types of stimuli, other than olfactory stimuli; in fact, this component is probably generated from particular areas of the nervous system (i.e., the hippocampus) that are not connected to a single human sense [84]. Furthermore, the correct interpretation of OERPs is still being debated. Generally, the OERPs components can be either present or absent; in some cases, it is present, but with altered characteristics in terms of latency and amplitude. However, normative data and their association with the severity of the olfactory disorders are not yet available [50]. In addition to this, establishing normative data for this electrophysiological measurement of olfactory dysfunction is also difficult since ERP is related to age, gender and personality, and these variables should be considered [72,84].

### 5.2. Improved Processing Techniques

Another concern is that OERP preprocessing is often based on grand averaging and, as aforementioned, this approach has some drawbacks. The main one is that the great amount of required stimuli is prohibitive [96,97]. This does not comply with the clinical tendency to minimize the duration and severity of the test and also to limit the phenomenon of habituation [98]. Therefore, a future step could be to investigate different approaches with specific regards to processing. The key issue would be to achieve a proper signal-to-noise ratio with a reduced number of trials. The literature suggests ERP analysis techniques based on single epoch analysis [63,65,99], improving the grand average with a priori knowledge [62,66], time–frequency analysis techniques (i.e., wavelet) [100,101], and Shannon entropy [102].

### 5.3. Machine Learning and Deep Learning

Finally, it is worth noting the growing need to identify and analyze alternative or integrative improvements of machine-learning approaches, in order to recognize features extracted from EEG signals recorded during perception of an odor and to diagnose any abnormalities in the patient. Indeed, in the literature, studies have been conducted for the implementation of classification methods useful to recognize signals belonging to the pleasantness of olfactory stimulation. In detail, in [103], a robust leave-one-subject-out classification method with a linear discriminant analysis (LDA) is applied to automatically classify the human EEG response related to pleasant or unpleasant olfactory stimulation classes. Moreover, a set of features extracted of the EEG signal acquired from lateral electrodes shows that right and left hemispheres perform differently in response to various smells. Other issues investigated in the literature explore the possibility of building classifiers for the detection of specific perceived odors from EEG signals so as to determine what odor was inhaled. In this regard, a pilot study conducted to introduce an olfactory stimulus decoder based on features related to the nonlinear behavior of EEG data has been introduced to classify, by LDA, olfactory stimuli with a public dataset consisting of five healthy male subjects [104]. Despite these efforts, unfortunately, is still difficult to obtain a classifier specifically for the diagnosis of olfactory dysfunctions, possibly related to silent diseases. In this context, greater efforts in the field of machine and deep learning need to be conducted. In fact, the most used pattern recognition approaches are based on principal component analysis (PCA) [105] or linear discriminant analysis (LDA) [106], while the available proposed signal processing are based on the extraction of features such as power spectral density (PSD), Higuchi’s fractal dimension (HFD) [107] or largest Lyapunov exponent (LLE). Taking into account the above, it is necessary to improve and enforce innovative artificial intelligence algorithms that can be used efficiently also on olfactory evoked potentials, with a view to developing systems that can accurately recognize olfactory abnormalities in patients under examination.

In this context, it would be interesting and extremely useful to develop a predictor for a particular disease (such as AD or PD) based on olfactory evoked potentials. Notably, having available data from OERP analysis of olfactory function of patients with a particular disease and healthy subject, a machine learning model could be trained. Extracting appropriate features from the OERP signal (such as entropic features, or information related to latency and amplitudes of a signal component, or a combination of both), classification of particular disease could be possible, separating people with normal olfactory function and olfactory dysfunction. Thus, the use of artificial intelligence in the context of the Health 4.0 era, applied to olfactory evoked potentials, could support diagnosis, as has already happened in other medical fields (see [108,109,110,111]).

## 6. Conclusions

In this paper, an overview on the use of OERPs signals for an objective evaluation of olfactory diseases is presented. This method is based on the analysis of the electrical potential elicited in a specific part of the brain, in response to an olfactory stimulus. This technique has numerous advantages over traditional techniques; in fact, the behavioral tests used in the past suffer from low clinical accuracy and are influenced by the subject’s state. Conversely, OERPs and the study of the neurocognitive response of the brain in a objective way would provide more reliable information on the health status of the subject/patient. For these reasons, this review highlights the innovative aspects of OERPs signals: starting from the description of the acquisition phase and of most adopted processing techniques, a critical survey of the most promising results in different application fields is reported. In particular, it was shown that OERP signals can be used to identify patients with incipient neurological diseases before the disease has taken over. Finally, the challenges and future goals were described since OERPs are expected to have a more significant future impact, especially as a support in clinical diagnostics. In this regard, the paper presents the main state-of-the-art contributions, useful for future research works in the field, especially in view of possible improvements and optimization of the existing methods for early stage diagnosis of smell disorders and for the prevention of several people with risk factors. The progress of OERPs signals is significant, although additional effort and improvement are required to meet the stringent requirements of clinical and diagnostic application areas.

## Figures and Tables

**Figure 1 bioengineering-09-00252-f001:**
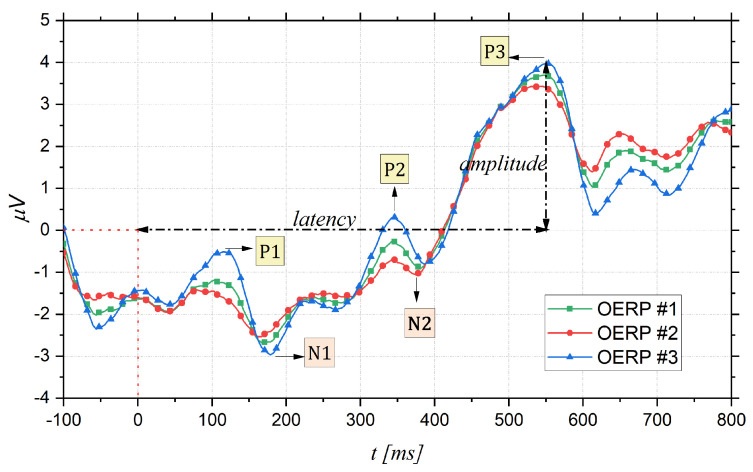
Examples of OERPs: peak amplitude is measured from baseline (0 μV) and peak latency from stimulus onset (0 ms).

**Figure 2 bioengineering-09-00252-f002:**
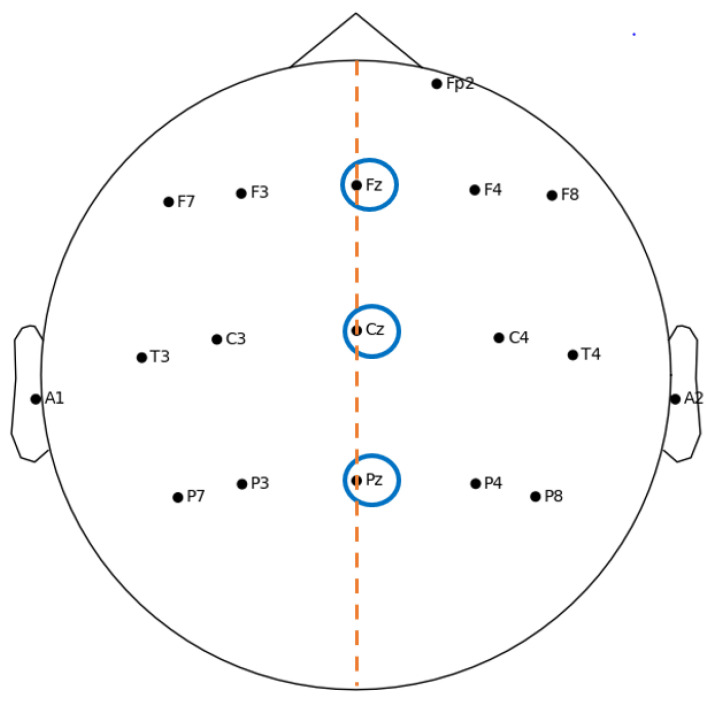
International 10–20 system (channel reduction). EEG activity related to olfactory is recorded with electrodes placed in the midline (orange line) at the Fz, Cz and Pz sites.

**Figure 3 bioengineering-09-00252-f003:**
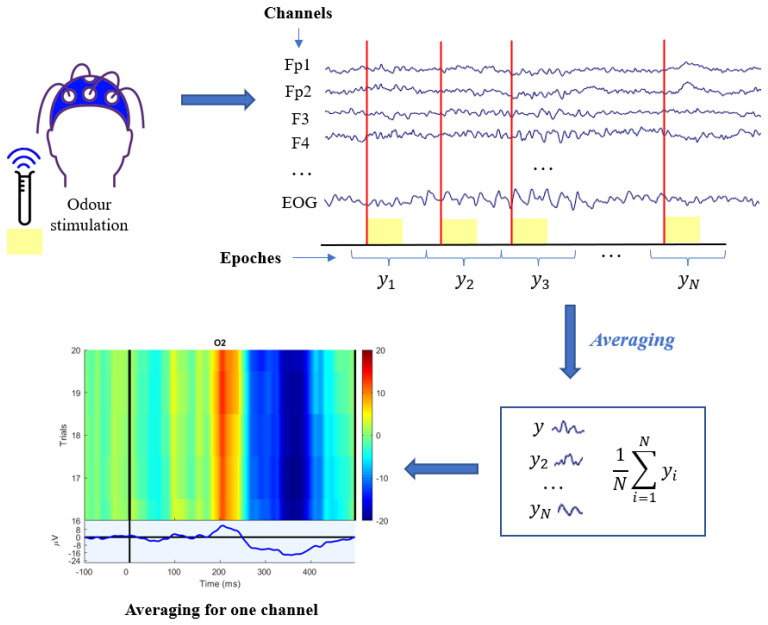
Grand averaging and its construction. In the ERP-image plot, each single trial is encoded as a colored line, warm colors representing positive activity and cool colors, negative activity.

**Figure 4 bioengineering-09-00252-f004:**
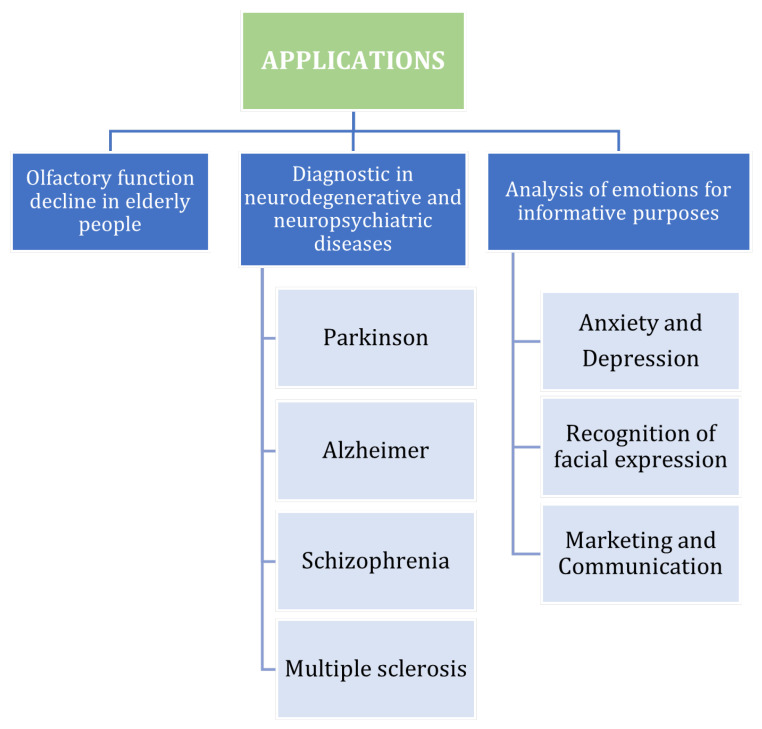
Taxonomy of the potential and application fields of OERP, resulting from the review of the literature.

**Figure 5 bioengineering-09-00252-f005:**
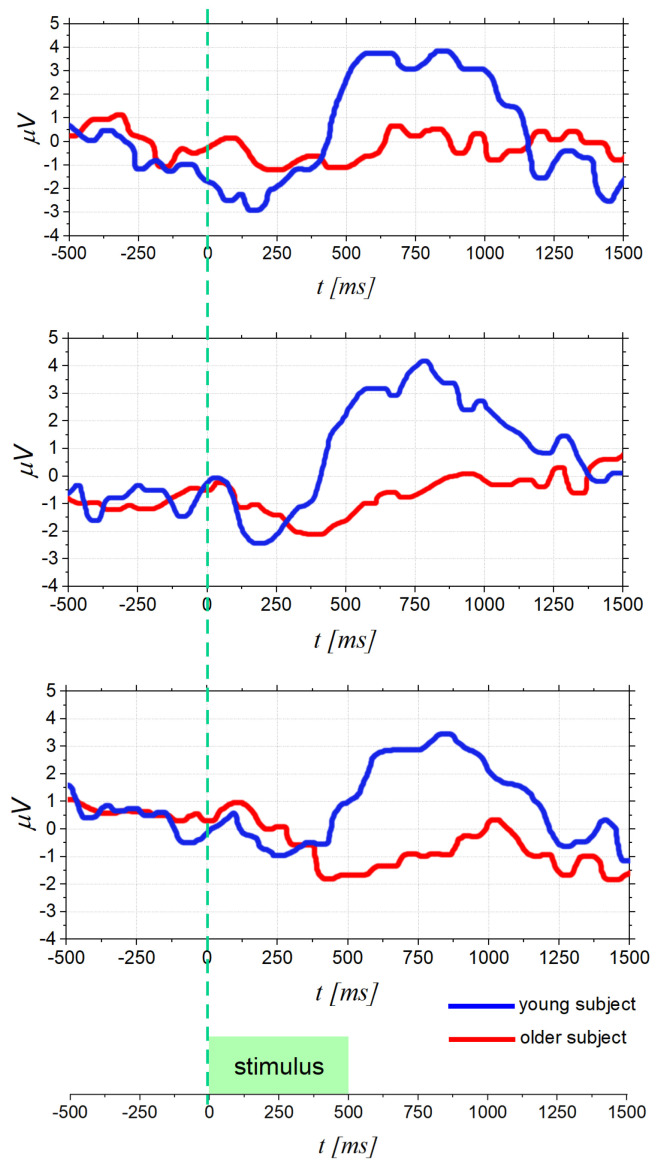
OERPsfor different examples of young and older subjects. OERP amplitude of an older participant is smaller and the peak latency is longer than that of a young participant [57].

**Figure 6 bioengineering-09-00252-f006:**
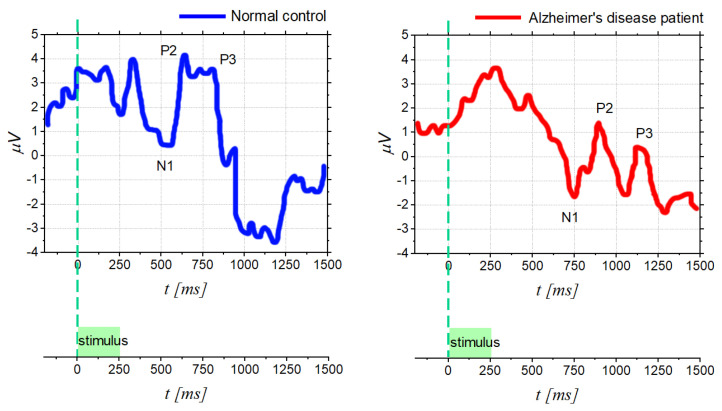
OERPfor a normal control subject and for a patient with Alzheimer’s disease (age 79), recorded at Pz position [83].

**Figure 7 bioengineering-09-00252-f007:**
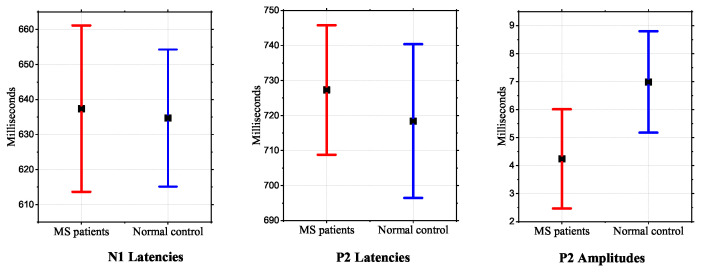
Comparison of the parameters of OERP components of multiple sclerosis patients and healthy controls in [85].

**Table 1 bioengineering-09-00252-t001:** Clinical assessment of olfactory function when combining the psychophysical and electrophysiological testing (OERPs) [50]. Copyright © 2007 The Triological Society Andrè Mouraux, Thomas Keller, Bernard Bertrand, et al., Clinical Significance of Olfactory Event-Related Potentials Related to Orthonasal and Retronasal Olfactory Testing, The Laryngoscope.

Psychophysical Orthonasal Testing (e.g., Sniffin’ Sticks)	OERP	Conclusion
Normosmia	Present	Normal olfactory function
Normosmia	Absent	Possibly normal olfactory function, consider the possibility of a technical problem (e.g., EEG artifacts)
Hyposmia	Present	Decreased olfactory function (the presence of OERPs may be correlated with a good prognosis)
Hyposmia	Absent	Decreased olfactory function (the absence of OERPs may be correlated with a poor prognosis)
Anosmia	Present	Consider patient malingering
Anosmia	Absent	Severely altered olfactory function, poor prognosis

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
