# Peer review of "Assessment and Scientific Progresses in the Analysis of Olfactory Evoked Potentials"

_bioengineering, 2022, doi:10.3390/bioengineering9060252_

Round 1
Reviewer 1 Report
Manuscript entitled "Assessment and Scientific Progresses in the Analysis of Olfactory Evoked Potentials" describes smell disorder diagnosis. The overall presentation of the manuscript is fine. Suggestions are as follows:
- Authors should cite related reviews to justify the novelty of the work.
- Mechanism of EEG signal generation and acquisition should be explained in detail. Authors should highlight hardware details as well.
- Manuscript lacks insights on data set generation.
- Machine learning and deep learning models should be explained.
- The suggestion for classification or prediction model for selective detection of a particular disease can be included in the revised version.
Reviewer 2 Report
The authors attempt to summarize the clinical use of olfactory event-related potentials. Unfortunately, the paper is difficult to follow. Topics are presented seemingly randomly, details are presented without context, and variables are mislabeled. Further the tone and tense of the article changes from sentence to sentence.
The paper suffers from a lack of synthesis. The presented citations are not made more clear by comparison or added context. The paper reads like a series of sentences where the reader does not gain added knowledge.
Some comments and edits are included.

Reviewer 3 Report
The authors review the development and the recent advances of olfactory event-related potentials (OERPs). The contents are well organized with major aspects of this technique.
The researchers in the olfactory sensor field are mostly not familiar with these OERPs/EEG fields, and this situation is the major problem for making a breakthrough in the sensor field. Thus, it would be interesting to include the potential applicability of these knowledge and techniques to the development of olfactory sensors, although this revision is optional.
Round 2
Reviewer 1 Report
Manuscript has been carefully revised and can be considered for publication.